# Performance of the METRIC model in estimating evapotranspiration fluxes over an irrigated field in Saudi Arabia using Landsat-8 images

Rangaswamy Madugundu[1], Khalid A. Al-Gaadi[1,2], ElKamil Tola[1], Abdalhaleem A. Hassaballa[1], Virupakshagouda C. Patil[1]

[1]Precision Agriculture Research Chair, King Saud University, Riyadh, 11451, Saudi Arabia
[2]Department of Agricultural Engineering, College of Food and Agriculture Sciences, King Saud University, Riyadh, 11451, Saudi Arabia

*Correspondence to*: Rangaswamy Madugundu (rmadugundu@ksu.edu.sa)

**Abstract.** Accurate estimation of evapotranspiration (ET) is essential for hydrological modelling and efficient crop water management in hyper-arid climates. In this study, we applied the METRIC algorithm was applied on Landsat-8 images, acquired from June to October 2013, for the mapping of ET of a 50-ha center pivot irrigated alfalfa field in the Eastern Region of Saudi Arabia. The METRIC estimated energy balance components and ET were evaluated against the data provided by an Eddy Covariance (EC) flux tower installed in the field. Results indicated that the METRIC algorithm provided accurate ET estimates over the study area, with RMSE values of 0.13 mm h$^{-1}$ and 4.15 mm d$^{-1}$. The METRIC algorithm was observed to perform better in full canopy conditions compared to that in partial canopy conditions. On average, the METRIC algorithm overestimated the hourly ET by 6.6 % in comparison to the EC measurements; however, the daily ET was underestimated by 4.2 %.

**Keywords.** Arid region, energy balance, eddy covariance, Landsat-8

## 1 Introduction

In the Kingdom of Saudi Arabia (KSA), the agricultural sector consumed about 85 % of the total freshwater used in 2008 (Al-Kahtani and Ismaiel, 2010). This share increased to 90 % by 2012 (Elnesr and Alazba, 2013). Hence, efficient use of water for crop production is essential to fulfilling the needs of the increasing population in KSA (Hussain et al., 2010; Praveen et al., 2012). Various studies (Kassem and Al-Moshileh, 2008; Atta et al., 2011; Al-Ghobari et al., 2013; Mohammad et al., 2013) recommend the development of advanced irrigation systems for improving agricultural water use efficiency in the Kingdom. As an example, a reduction in the irrigation water of 30-40 % could be attained when sprinkler irrigation is used instead of traditional methods. An additional saving of 10-25 % can be reached with drip irrigation systems (Rizaiza and Al-Osaimy, 1996). Furthermore, implementation of recent innovative precision irrigation technologies, in conjunction with the accurate estimation of crop water requirements through remotely sensed data, could significantly enhance the efficient use of irrigation water in the agricultural sector.

Evapotranspiration (ET) measurements play a crucial role for water management under hyper-arid conditions. Particularly, in irrigation scheduling, hydrologic modelling and drought monitoring (Bastiaanssen et al., 2000; Allen et al., 2005; Chavez et al., 2005; Senay et al., 2008; Santos et al., 2010; Anderson et al., 2012; Mkhwanazi and Chavez, 2012; Gowda et al.,2013; Lagos et al., 2013; Moorhead et al., 2013; Yee et al., 2014; Madugundu et al., 2017). In semi-arid climates, ET is characterised as one of the most significant components influencing the hydrologic cycle. Hence, accurate determination of ET is considered as one of the crucial/essential factors influencing the optimal management of crop water use through different irrigation systems (Hoedjes et al., 2008). Also, the accurate determination of the energy balance (EB) components such as sensible heat (H), soil heat (G) and latent heat (LE) fluxes, is of great value for water management practices, in these arid and semi-arid regions (Zeweldi et al., 2010).

The ET can be estimated with different direct/indirect methods such as the lysimeter, the water balance, the Bowen Ratio (BR), the Eddy Covariance (EC) and the scintillometer (SC) method (Allen et al., 2011a; Rana and Katerji, 2000). Lysimeter and EC methods provide direct measurements of ET, while other methods rely on models to estimate heat fluxes using measurable parameters (FAO, 1977; Drexler et al., 2004; Chavez et al., 2009a). The FAO-56 (Allen et al., 2007) modelling for instance, is widely used in numerous studies. This method consists of estimating crop evapotranspiration (ETc) for a crop canopy using a reference evapotranspiration (ETr) and a crop coefficient (Kc). The ETr is computed based on the Penman-Monteith (PM) method. The FAO-56 computed actual ET can be used to monitor spatially distributed ET developed with RS approaches (Courault et al., 2005).

Recently, EC systems have gained a lot of popularity in the determination of ET. Several studies have been conducted to investigate the effectiveness of the EC systems in estimating accurate values of H, LE and ET. Yet, the method is often constrained by the lack of surface energy budget closure. For example, Chavez et al. (2009b) reported that the EC measured energy balance components (H and LE) underestimated about 30 % of these components as inferred from the large weighing lysimeter on an irrigated cotton field in the USA. Similarly, Twine et al. (2000) reported that carbon dioxide fluxes measured over a sorghum crop with four EC systems (equipped with the same models of instruments), underestimated the reference values with the same factor when energy balance closure was not achieved. However, the EB closure can be improved by adjusting the EC data through the Bowen ratio-BR (Ding et al., 2010; Teixeira and Bastiaanssen, 2012). Ding et al. (2010) reported that the EC measured ET (ET$_{EC}$) compared to the lysimetric data (ET$_L$) of maize crop showed an EB closure of up to 84 % for the daytime fluxes. However, the forced energy balance closure with BR data improved the EB closure from 79.2 % to 95.2 %. Therefore, adjusting ET values measured by the EC system, using the BR method improves the accuracy.

In-situ point measurements provide accurate ET values; however, these methods are limited to small areas only. Alternative method such as remote sensing (RS) methods, have been successfully used for assessing the spatial distribution of ET on larger scales, over different landscapes (Bala et al., 2013; Farah et al., 2004; Colaizzi et al., 2006) and agricultural fields (Kalma et al., 2008; Gowda et al., 2008). One of the most commonly used RS methods for the determination of ET is the Surface Energy Balance Algorithm for Land (SEBAL) method. This method has been demonstrated to assess ET with an accuracy ranging from 67 % to 97 % (Bastiaanssen et al., 2005). Recently, several SEBAL versions have been developed and

successfully implemented in different water management practices (Paul et al., 2013). Examples of such methods include the Mapping Evapotranspiration at high Resolution and with Internalized Calibration (METRIC), the Modified SEBAL (M-SEBAL) method, the Simplified Surface Energy Balance (SSEB) method, the Remote Sensing of Evapotranspiration (ReSET) method, the Surface Energy Balance System (SEBS) method and the Surface Energy Balance with Topography Algorithm (SEBTA) method. The METRIC method, which was developed to determine the quantity and spatial distribution of ET over large areas, is considered as one of the most appropriate models for the continuous estimation of ET over crops during the growing season (Allen et al., 2007; Patil et al., 2015).

Over the past decades, several hydrological studies in the KSA have been directed to accurate RS based ET estimates at various spatial and temporal scales. As an example, Mahmoud and Alazba (2016) processed a spatial ET dataset over Western and Southern regions of Saudi Arabia with SEBAL on MODIS. However, the quantification of ET from satellite data in conjunction with EC data is still limited in the KSA. As the Eastern Region in the KSA experiences severe dry conditions, long-term EC data in conjunction with Landsat-8 data will enhance the accurate estimates and distribution of ET over various landscapes/agricultural fields. In view of the pressing need to assess the productivity of agricultural fields in the KSA, this study was undertaken in an attempt of applying the METRIC model on Landsat-8 imagery for the determination of spatial and temporal variability in ET aiming at optimizing the quantification of crop water requirement and the formulation of efficient irrigation schedules.

## 2 Materials and Methods

### 2.1 Study area

The study was carried out on a 50-ha alfalfa field, which was one of the 48 agricultural fields of Todhia Arable Farm (TAF) located about 250 km Southeast of Riyadh, the capital city of Saudi Arabia, at coordinates of 24° 11' 00" E and 48° 56' 14.6" N (**Fig. 1**). The farm was under an arid climate with hot summers (40 ± 2 °C) and cold to moderate winters (15 ± 3 °C) and a mean air temperature of 35°C. The annual rainfall was about 90 mm, most of which occurred in the period from November to February. The soil of the study area is sandy loam soil with a mean value of soil pH of 7.58 and soil electrical conductivity (EC) of 2.36 dS m$^{-1}$. The field is a flat terrain with slight undulations in the desert environment with an elevation ranging between 329 m and 453 m. The major crops cultivated in the study area were alfalfa (*Medicago sativa* L.), Rhodes grass (*Chloris gayana*. Kunth) and corn (*Zea mays*. L). Due to the high crop water demand combined with the highly erratic rainfall, irrigation is a pre-requisite for crop growth. It is entirely provided using groundwater delivered by center pivot irrigation systems. The experimental field was cultivated with an alfalfa crop (Green Master) sown on December 6$^{th}$, 2012 at a seeding rate of 20 kg ha$^{-1}$, and was irrigated through a center pivot system using a groundwater of mean values of EC, pH and Sodium Absorption Ratio of 2.917 dS m$^{-1}$, 7.82 and 1.42, respectively. In the study region, alfalfa crop is usually cultivated for two years; with, on the average, a cutting each 30-35 days during summer periods and every 45-60 days cut during winter periods.

The mean alfalfa hay yield for each cut is estimated to be 2.0-3.0 kg ha$^{-1}$ during winter and 4.0-5.0 kg ha$^{-1}$ during spring/summer periods (Kayad et al., 2016).

## 2.2 Eddy Covariance (EC) flux data

The Eddy Covariance (EC) system was installed, on May 27$^{th}$, 2013, over the selected alfalfa field. The EC flux tower was powered by solar panels with rechargeable batteries. The meteorological and gas exchange (flux) measurements were made at a height of 3.67 m. The EC was equipped with response sensors, including an open-path infrared gas analyzer, IRGA (LI-7500), a 3-axis ultrasonic anemometer (3D Master Pro), Gill Instruments™, a net radiometer (CNR-4, Kipp and Zonen™), and a quantum sensor. Soil heat flux plates (HFP01, Hukseflux™) were installed at various depths (5, 10, 15, 20, 25 and 30 cm). The sonic anemometer and IRGA were set to record flux data logged at 10 Hz (CR3000) and stored as 30 min files (*.ghg). Data on crop condition, growth, and phenological parameters were recorded during the frequent field visits.

The recorded EC data, for the period from June to October 2013, was used for this study. The collected 30-minute raw datasets (*.ghg files) were processed with the "advanced mode" of an automated software program Eddypro (ver. 5.0) developed by LI-COR Biosciences, USA. With the Eddypro, wind speed measurement offsets are processed, axis rotation for tilt is corrected, de-trending is made for turbulent fluctuations, the time lag is optimized, spectral corrections and footprint estimation are made following the procedures described in the Eddypro software instruction manual (LI-COR). As the sonic anemometer was tilted 243º towards the North, an angle-of-attack correction for wind components was also performed. During the filtering process, EC data were excluded whenever there was a condensation, a covariance with missing values (-9999) or periods with incorrect sonic temperatures (i.e. >50 °C). The missing data were filled with the standard methods (basic interpolation) as described in the manual (LI-COR). The missing were linearly interpolated when gaps are no longer than an hour, whereas gaps due to a malfunction of EC system for longer than 24 hours were not removed. Subsequently, the processed EC data were used for the computation of the ET (i.e. ET$_{EC}$).

## 2.3 Footprint analysis (FTP)

As described in Nappo et al. (1982), the footprint (FTP) is "the extent to which a set of measurements was taken in a given space-time domain". In this study, an effort was made to understand the spatial representation area or footprint (FTP) of flux measurements from the EC system. The FTP mainly depends on the function of surface roughness with respect to the wind speed, direction, and wind shear. Thus, the FTP for a single 30-minute data record will be unique and may vary for the next segment as atmospheric conditions change. Moreover, the shape and length of the FTP may vary with upwind direction, as well as the relative weights (magnitude of flux contributions); in each area inside the FTP, i.e. the weighted FTP function is used to attribute the measured flux to a weighted areal estimate (Gockede et al., 2006). The areas very close to the EC tower may contribute less to the total flux sensed by the instrument. Whereas, areas away (upwind) from the EC tower contribute significantly, up to a point where a peak is reached. Thereafter, the contribution may decrease rapidly (Zhao et al, 2014).

Depending on the complexity, numerous FTP estimation models have been presented in the literature. The widely used Flux Source Area Model (FSAM) described by Schmid (1994), was used in this study for the determination of FTP. The FSAM was computed based on contributes (up to 90%) of the sensed fluxes by the EC tower. Subsequently, the obtained FTP was used to compare the EC tower and Landsat-8 estimates of energy fluxes. As described by Schmid (1994), the EC contributions starting 1.5 hours after sunrise and ending 1.5 hours before sunset was considered and the FTP was computed for every 30 minutes corresponding to the averaging period used for the EC tower analysis. The FSAM variables such as observation point height, Obukhov length, the standard deviation of lateral wind speed fluctuations; and friction velocity were taken from the EC measurements. Surface roughness length was estimated by measuring the crop (alfalfa) height at regular site visits using linear interpolation between observations. Crop height was transformed into roughness length as described by Allen et al. (1998).

In the FTP analysis, the utilized wind parameters such as wind speed, wind direction, and yaw angle at the time of Landsat-8 overpass were provided in **Table 1**. The produced oval-shaped "fetch area" was oriented to the mean wind direction corresponding to that half hour time-period. A weighting matrix for the FTP surrounding the EC tower was developed for each day. Subsequently, a 30-meter grid matching to the Landsat-8 pixels was created for each 30-minute interval, with a binary coding. The grid cell which falls "in the fetch area" was coded as "1", and the pixels "outside the fetch area" were given the value of "0". The daily FTP fetch calculation was computed by scoring the grid cells coded with "1" divided by the total number of observations recorded during the day.

### 2.4 Energy balance (EB)

The Eddy Covariance (EC) system allows accurate measurements of the energy balance (EB) components. Hence, using the measured EB components, the surface energy budget can be estimated (Equation 1) along with the EB closure. The four flux components ($R_n$, H, G and LE) measured by the EC system were assessed for the EB closure (Kustas et al., 2005). The EB closure rate for the EC system ranged between 71% and 99%, with a mean value of 87% **(Table 2 and Fig. 2).** The Energy balance closure was significantly influenced by the amount of the unaccounted energy lost (i.e. lack of closure in the energy balance equation due to advection and measurement errors in the field) in the partition of the net radiation into latent and sensible heat fluxes and soil heat flux. It may be due to the portion of energy that is missing in the budget. It was accounted for between 5 and 20 % of the total $R_n$. This was attributed to the inherent errors in measuring the EB components by the ground systems, which was reported to be 15 – 20 %, 5 – 10 % and 20 – 30 % according to Weaver (1990), Field et al. (1994) and Twine et al. (2000), respectively.

### 2.5 Landsat-8 images and pre-processing

Eight cloud-free Landsat-8 TIRS (Thermal Infrared Sensor) and OLI (Operational Land Imaginer) data of experimental field (Path 165, Row 43) were downloaded for the study period (June to October 2013) from the United States Geological Survey

(USGS) Earth Explorer site (http://earthexplorer.usgs.gov), **Table 3**. The downloaded data were georeferenced to the Universal Transverse Mercator (UTM) Map projection using the World Geodetic System 84 (WGS84) datum. Subsequently, Fast Line-of-sight Atmospheric Analysis of Hypercubes (FLAASH) algorithm in ENVI software was used to convert Landsat-8 digital numbers to Top-Of-Atmosphere (TOA) spectral reflectance. Also, land surface temperature (LST) layers were generated using split window algorithm (USGS). The obtained spectral reflectance and the land surface temperature values were used as inputs for the ET estimation employing the METRIC algorithm. Image analysis and the execution of the METRIC algorithm were performed using ENVI software program (ver. 5.1). Subsequently, spatially distributed ET fluxes were weighted and integrated using an FTP for comparison with to ET measured with EC system.

**2.6 Land Surface Temperature (LST) – Split window (SW) algorithm**

Landsat-8 TIRS Bands (10 and 11) and OLI Bands (2 to 5) were utilized in the estimation of LST through Split Window algorithm executed in ENVI (ver. 5.1). During the process of LST estimation, the algorithm used brightness temperatures (TB) of two TIRS bands, and mean and the difference in land surface emissivity (LSE) of an area under observation. The SW algorithm for LST determination was given in Equation (1) as described by Skokovic et al., (2014).

$$LST = TB_{10} + C_1(TB_{10} - TB_{11}) + C_2(TB_{10} - TB_{11})^2 + C_0 + (C_3 + C_4 W)(1 - \varepsilon) + (C_5 + C_6 W)\Delta\varepsilon \qquad (1)$$

where, LST is the Land Surface Temperature (K); $C_0$ to $C_6$, are the values of Split-Window coefficient (**Table 4**); $TB_{10}$ and $TB_{11}$ are brightness temperatures of Landsat-8 band 10 and 11 (K); $\varepsilon$ is the mean LSE of TIR bands; W is the atmospheric water vapour content; and $\Delta\varepsilon$ is the difference in LSE.

The brightness temperature (TB), a microwave radiation radiance travelling upward from the top of Earth's atmosphere, was calculated using Equation (2).

$$TB = \frac{K_2}{Ln\left(1 + \frac{K_1}{L\lambda}\right)} \qquad (2)$$

where, $K_1$ and $K_2$ are thermal conversion constants of TIRS bands (**Table 5**) and $L\lambda$ - is the Top of Atmospheric spectral radiance. This Top of Atmosphere spectral radiance was determined by multiplying multiplicative rescaling factors (**Table 5**) from Equation (3).

$$L\lambda = M_L \times Q_{cal} + A_L \qquad (3)$$

where, $M_L$ is the band specific multiplicative rescaling factor (radiance_mult_band_10/11); $Q_{cal}$ – is the band 10/11 image; $A_L$ is the band specific additive rescaling factors (radiance_add_band_10/11).

Subsequently, LSE was calculated using Equation (4). The $\varepsilon_s$ and $\varepsilon_v$ are soil and vegetation emissivity values of the corresponding bands (**Table 5**). The fractional vegetation cover (FVC) was estimated based on the Normalized Difference Vegetation Index (NDVI) obtained over the experimental area (Equation 5).

5  $LSE = \varepsilon_s(1 - FVC) + \varepsilon_v * FVC$   (4)

$FVC = \dfrac{NDVI - NDVI_s}{NDVI_v - NDVI_s}$   (5)

where, $NDVI_S$ and $NDVI_V$ are the reclassified NDVI for soil and vegetation areas respectively. OLI bands 2, 3, 4 and 5 were layer stacked and NDVI was calculated using ENVI (ver. 5.1) software. The output value of NDVI ranged between -1 and 10   0.59. To get $NDVI_S$ and $NDVI_V$, the NDVI image was reclassified into soil and vegetation. After generating LSE for both the bands of TIR, the mean and difference LSE were obtained according to Equations (6 and 7).

$\varepsilon = \dfrac{\varepsilon_{10} - \varepsilon_{11}}{2}$   (6)

15   $\Delta\varepsilon = \varepsilon_{10} - \varepsilon_{11}$   (7)

### 2.7 METRIC algorithm and ET estimation

The METRIC algorithm has been developed exclusively for the estimation of ET from Landsat data (Allen et al., 2005). A 20   total of eight Landsat-8 images were processed and used for the estimation of ET ($ET_{METRIC}$) over the experimental alfalfa field. During the $ET_{METRIC}$ computation, surface characteristics such as surface albedo, vegetation indices, surface emissivity and surface temperature were estimated as intermediate products. Anchor pixels (hot and cold) were selected, and the energy components such as the net radiation ($R_n$), the soil heat flux (G) and the sensible heat flux (H) were estimated as well. Finally, the latent heat flux (LE) was predicted as a residual of the land surface balance (Allen et al., 2005, 2007), Equation (8). 25   Consequently, the instantaneous ET ($ET_{inst}$) for each pixel was calculated. Moreover, the obtained leaf area index ($LAI_G$) and the reference ET (ETr) over the alfalfa field were used as inputs to the METRIC-based Landsat-8 ET prediction (**Table 6**). The ground-based LAI ($LAI_G$) of alfalfa was recorded on the day of satellite overpass using a plant canopy analyzer (LAI-2200).

30   $LE = R_n - G - H$   (8)

The first step in the METRIC model was to compute the net radiation ($R_n$) using the surface radiation balance, Equation (9). The $R_n$ estimation was accomplished in a series of steps by summing up the net short-wave radiation and net long-wave radiation (Hipps, 1989; Brunsell and Gillies, 2002; Allen et al., 2007).

$$R_n = R_{s\downarrow} - \alpha\, R_{s\downarrow} - R_{L\downarrow} - R_{L\uparrow} - (1 - \varepsilon_o)\, R_{L\downarrow} \qquad (9)$$

where, $R_{S\downarrow}$ is the incoming shortwave radiation (W m$^{-2}$); $\alpha$ is the broadband surface albedo (dimensionless); $R_{L\downarrow}$ and $R_{L\uparrow}$ are the inbound and outgoing longwave radiation (W m$^{-2}$), respectively. $\varepsilon_o$ is the broadband surface thermal emissivity (dimensionless). The $(1-\varepsilon_o)\, R_{L\downarrow}$ term represents the fraction of incoming longwave radiation reflected from the surface.

The incoming broadband and shortwave radiation ($R_{S\downarrow}$), which represents the principal energy source for ET, is calculated for the Landsat-8 image time as a constant for the whole image assuming clear sky conditions as per Equation (10).

$$R_{S\downarrow} = \frac{G_{SC}\, cos\theta_{ref}\, \tau_{sw}}{d^2} \qquad (10)$$

where, $G_{SC}$ is the solar constant (1367 W m$^{-2}$); $\theta_{ref}$ is the solar incidence angle; $\tau_{sw}$ is the broadband atmospheric transmissivity and $d^2$ is the square of relative Earth-Sun distance. The $\tau_{sw}$ is calculated using Equation (11) drafted in the ASCE-EWRI (2005).

$$\tau_{sw} = 0.35 + 0.627\, exp\left[\frac{-0.00146P}{K_t \cos Z} - 0.075\left(\frac{W}{\cos Z}\right)^{0.4}\right] \qquad (11)$$

where, P is the atmospheric pressure (kPa); W is the amount of water present in the atmosphere (mm); Z is the solar zenith angle (extracted from the image metadata); and $K_t$ is air turbidity coefficient ($K_t = 1.0$ for clean air and 0.5 for extremely turbid or polluted air. $K_t = 1.0$ was used in this study). The P and W are calculated, using the measured/estimated near-surface vapour pressure, as per equations (12 and 13) according to ASCE-EWRI (2005) and Garrison and Adler (1990), respectively.

$$P = 101.3 \left(\frac{293 - 0.0065_z}{293}\right)^{5.26} \qquad (12)$$

$$W = 0.14 e_a\, P_{air} + 2.1 \qquad (13)$$

where, the constant 293 is the standard air temperature (K); z is the elevation above the sea level (m) and the $e_a$ is the near-
30 surface vapour pressure (kPa). The parameter $d^2$ was computed, from Equation (14), as a function of the day of the year (DOY) as described in Duffie and Beckman (2013).

$$d^2 = \frac{1}{1 + 0.033 \cos(DOY\, 2\pi/365)} \qquad (14)$$

The broadband surface albedo ($\alpha$), however, is calculated using Equation (15) as described in Zhong and Li (1988) and Bastiaanssen et al. (1998).

$$\alpha = \frac{(\alpha_{toa} - \alpha_{atm})}{\tau_{sw}^2} \qquad (15)$$

where, $\alpha_{toa}$ is the planetary albedo of each pixel; $\alpha_{atm}$ atmospheric albedo and $\tau_{sw}$ is obtained from the Equation (11) following Silva et al. (2016).

Outgoing longwave radiation ($R_L\uparrow$) emitted from the surface is driven by surface temperature and surface emissivity. The $R_L\uparrow$ is computed using the Stefan-Boltzmann equation (16).

$$R_{L\downarrow} = \varepsilon_o \sigma T_s^4 \qquad (16)$$

where $\varepsilon_o$ is the broadband surface emissivity (dimensionless); $\sigma$ is the Stefan-Boltzmann constant ($5.67 \times 10^{-8}$ W m$^{-2}$ K$^{-4}$); and Ts is the surface temperature (K). In this study, the $T_s$ was accounted as LST and obtained from Equation (1). The surface emissivity was computed using an empirical Equation (17) after Tasumi (2008) based on soil and vegetative thermal emissivities. The leaf area index (LAI) is computed as per Equation (18) proposed by Bastiaanssen (1998). During the correction, the typical bare soil and the fully vegetated surface values were set as 0.93 and 0.98, respectively. The soil-adjusted vegetation index (SAVI) was calculated based on TOA reflectance of bands 4 and 5 (Huete, 1988).

$$\varepsilon_o = 0.95 + 0.01\, LAI \; for \; LAI \leq 3 \qquad (17)$$

$$LAI = \frac{-\ln[(0.69 - SAVI)/0.59]}{0.91} \qquad (18)$$

The incoming longwave radiation ($R_L\downarrow$); a downward thermal radiation flux from the atmosphere (W m$^{-2}$); was estimated using the Stefan-Boltzmann Equation (19) described in Allen et al. (2007).

$$R_{L\downarrow} = \varepsilon_a \sigma T_s^4 \qquad (19)$$

where, $\varepsilon_a$ is the broadband surface emissivity (dimensionless); $\sigma$ is the Stefan-Boltzmann constant ($5.67 \times 10^{-8}$ W m$^{-2}$ K$^{-4}$); and $T_s$ is the surface temperature (K). The $\varepsilon_a$ was computed using Equation (20) as described in Bastiaanssen (1995) and Allen et al. (2000).

$$\varepsilon_a = 0.85 \, (-ln \, \tau_{sw})^{0.09} \qquad (20)$$

where, $\tau_{sw}$ is the broadband atmospheric transmissivity calculated from Equation (11).

For the estimation of soil heat flux (G), various empirical equations can be found in the literature (Bastiaanssen, 1998; Singh et al., 2008; Gowda et al., 2011). However, this study adopted the empirical model described by Bastiaanssen (2000) representing values near midday for the prediction of Landsat-8 G (i.e. $G_{METRIC}$), as a ratio $G/R_n$ based on the NDVI, Equation (21).

$$\frac{G}{R_n} = (T_s - 273.15)(0.0038 + 0.0074 \propto)(1 - 0.98 \, NDVI^4) \qquad (21)$$

where, $T_s$ is the surface temperature (K) and $\alpha$ is the surface albedo. Subsequently, the $G_{METRIC}$ was obtained by multiplying $G/R_n$ with $R_n$.

The sensible heat ($H_{METRIC}$) was estimated from an aerodynamic function as expressed in Equation (22). In the calculation of $r_{ah}$, wind speed measurements were used.

$$H = \rho_{air} \, C_p \, \frac{\Delta T}{r_{ah}} \qquad (22)$$

where $\rho$ is the air density (kg m$^{-3}$); $C_p$ is the specific heat capacity of the air (J kg$^{-1}$ $^\circ$K$^{-1}$); $\Delta T$ is the near-surface air temperature gradient and $r_{ah}$ is the aerodynamic resistance for heat transfer (s m$^{-1}$) between two near-surface heights (i.e. at alfalfa canopy height and the EC measurement height). For further details of the $H_{METRIC}$ algorithm, we refer to Allen et al. (2007).

**2.8 Calculation of ET**

Following the establishment of $R_n$, G and H from the Landsat-8 processing, LE was calculated as a residual of the energy balance equation. The obtained LE is equivalent to the $ET_{inst}$ at the time of Landsat-8 overpass, Equation (23).

$$ET_{inst} = 3,600 \frac{LE}{\lambda \rho_\omega} \quad (23)$$

where, $ET_{inst}$ is the instantaneous ET (mm h$^{-1}$); 3,600 converts from seconds to hours; $\rho_\omega$ is the density of water ($\sim$1,000 kg m$^{-3}$) and $\lambda$ is the latent heat of vaporization (J kg$^{-1}$) representing the heat absorbed when a kilogram of water evaporates. The $\lambda$ component was computed as per Equation (24).

$$\lambda = [2.501 - 0.00236\ (T_s - 273.15)] \times 10^6 \quad (24)$$

Finally, as presented in Equation (25), the reference ET fraction (ET$_r$F) was calculated as the ratio of the computed ET$_{inst}$ from each pixel to the reference ET (ET$_r$) computed from the weather data.

$$ET_rF = \frac{ET_{inst}}{ET_r} \quad (25)$$

where, ET$_{inst}$ is from Equation (23) and ET$_r$ is for the standardized 0.5 m tall alfalfa at the time of the image. The EC system recorded weather parameters were used to calculate ET$_r$ as described in ASCE-EWRI (2005). The obtained ET$_r$F was subsequently extrapolated to daily values. In the processes, ET$_{24}$ was computed by assuming that the instantaneous ET$_r$F computed at satellite overpass is the same as the average ET$_r$F over the 24 h average (Allen et al., 2007), Equation (26).

$$ET_{24} = C_{rad}\ (EF)(ET_{r24}) \quad (26)$$

**2.9 Data analysis**

The spatially estimated energy flux components, derived from the METRIC algorithm and the EC flux tower, were subjected to heat flux correction and energy balance and footprint (FTP) analysis, as described in Schmid (1994). Different statistical performance indicators (RMSE, MBE, and Nash-Sutcliff coefficient). As the samples were limited, the Mann-Whitney U test and/or Kruskal-Wallis H test (Gisondi et al., 2004; McCune and Grace, 2002) were performed for the assessment of the METRIC performance in estimating ET against the EC system. These statistical indicators are often used when small sample sizes are considered.

# 3 Results and discussion

## 3.1 Footprint analysis (FTP)

The simple arithmetic averages of weighted/integrated heat fluxes over the fetch areas are the FTPs (Chavez et al., 2005). FTPs are widely used in validating the spatially distributed fluxes obtained from remote sensing (RS) with EC measured fluxes. The fetch area of the FTP was classified into six classes, based on the cumulative contribution of the eddies, as provided in **Fig. 3**. Based on the FTP analysis, about 90 % of the EC system observes eddies originating from the 122 to 144 m in the upwind direction. The peak (vertical) fetch is within 42 to 51 m. It is also observed that 10 % of the fluxes are registered between 6 and 17.5 m from the flux tower. About 30 %, 50 % and 70 % of the fluxes are recorded between 35 to 63 m, 52 to 97 m and 81 to 136 m, respectively. Energy fluxes are analyzed as shown by Chavez et al. (2005) by integrating Landsat-8 image acquisition (i.e. overpass). This allows the EC footprint to overlap the METRIC footprint by 90% (Kustas et al., 2005).

## 3.2 Surface temperature (T)

The linear regression analysis of Landsat-8 derived surface temperature ($T_{LST}$) against the EC flux tower measured temperature ($T_{EC}$), obtained from the upwind longwave component measured by the CNR4, shows a good correlation with an $R^2$ value of 0.71 (p = 0.0084) (**Fig. 4**). However, the $T_{LST}$ is underestimated compared to the $T_{EC}$ as evidenced by the RMSE and MBE performance indicators of 4.23 °C (-12.82 %) and -3.40 °C (-9.34 %), respectively. The recorded errors (4.2 % to 19.6 %) in the $T_{LST}$ are slightly higher than the values reported in previous studies. For example, compared to the ground level infrared thermometer, Chavez et al. (2009a) reported METRIC errors of 11.1 % and 1.9 % in estimating surface temperatures for corn and sorghum fields, respectively. The LAI significantly affects the accuracy of $T_{LST.}$ Vegetation with higher LAI records lower temperatures because the amount of heat stored is reduced through transpiration (Omran, 2012). In this study, the Landsat-8 derived $T_{LST}$ is underestimated by about 20 % at full foliage cover of alfalfa crop (i.e. LAI ~ 6) compared to $T_{EC}$. High LAI surfaces can trigger large coherent eddies that are efficient in heat convection; whereas, low LAI surfaces are less efficient in generating energetic eddies (Voogt and Grimmond, 2000). Thus, the temporal variation in LAI would result in oscillations of the land surface temperature. As an example, in the case of low LAI (<1.23) conditions, the $T_{LST}$ was about 40.99 °C; whereas, at higher LAI (5.92) the $T_{LST}$ was 32.6 °C. However, there is a discrepancy at higher temperatures when the crop density of alfalfa was low (i.e. bare soil was visible to the radiometer) as evidenced by the low LAI values over the footprint and the recorded $T_{EC}$ and $T_{LST}$ (Chavez et al., 2005). Hence, the cooling effect of vegetation on the $T_{LST}$ accelerated the error by 1 to 20 % on both the low LAI and high LAI conditions.

## 3.3 Energy balance (EB) components

The mean values of energy balance (EB) components obtained from both the EC system and METRIC algorithm are provided in **Table 7**. While, the performance indicators (RMSE, MBE, Nash-Sutcliff coefficient, Mann-Whitney U-test and Kruskal-

Wallis H-test) are given in **Table 8**. An illustration (**Fig. 5**) on the comparison of EC and METRIC estimates of EB components are provided.

### 3.3.1 Net radiation ($R_n$)

The temporal trend of EC (CNR-4) measured $R_n$ ($R_{nEC}$) was analyzed for discrepancies. About 9 % to 16 % of the collected data exhibit infrequent errors. Hence, the abnormal data have been discarded, and a gap-filling process was performed using the Eddy Pro software program (version 5.0). The scatter plot (**Fig. 5**) represents the relationship between $R_{nEC}$ and METRIC estimated $R_n$ ($R_{nRS}$). The moderate linear correlation ($R^2 = 0.54$ and p>F = 0.0381), low RMSE (18.32 W m$^{-2}$; 3.8 %) and very low MBE (8.66 W m$^{-2}$; 1.76 %) indicate that the METRIC model accurately (96 %) estimate the $R_n$. The obtained RMSE and MBE values are in agreement with those reported by earlier studies of Mkhwanazi and Chavez (2012) where the RMSE value was 4.1 % and MBE value was 3.3 %, and Chavez et al. (2007) where the RMSE value was 9.8 %. In addition, the performance of METRIC was significant with a Kruskal-Wallis H-tests ($\chi^2 = 0.893$). However, the null hypothesis cannot be rejected with the Mann-Whitney U-test (U and Z value are 23 and -0.892 respectively). These variations in $R_n$ between METRIC and CNR-4 estimates are likely due to variations in the computation of leaf area index along with LST and soil moisture conditions (Zhang et al., 2013).

### 3.3.2 Soil heat flux (G)

**Figure 5** illustrates a scatter plot between the EC measured ($G_{EC}$) and the METRIC estimated ($G_{RS}$) soil heat flux values. The relationship was relatively fair with an $R^2$ value of 0.67 and a Nash-Sutcliffe coefficient of 0.59. The RMSE and MBE, however, were determined at 28.46 W m$^{-2}$ (37.33%) and 12.42 W m$^{-2}$ (16.29%), respectively. These recorded errors were relatively higher than the values reported in similar previous studies. Mkhwanazi and Chavez (2012) reported an RMSE value of 14.2 W m$^{-2}$ (27.6%) and an MBE value of -3.0 W m$^{-2}$ (-5.8%) in estimating G with the METRIC algorithm for irrigated alfalfa. The average observed values of G were similar to the majority of values recorded by Ham et al. (1991). The performance of METRIC model in estimating G was not significant with the Mann-Whitney U-test (U and Z value are 23 and -0.892 respectively) but it was significant with Kruskal-Wallis H-tests ($\chi^2 = 0.397$) and the null hypothesis was not rejected. Furthermore, the ratio of $R_n$ to the G (G/$R_n$) derived from METRIC model against the LAI inferred from the Canopy analyzer (PCA 2200, Li-COR, USA) is presented in **Fig. 6**. The G/$R_n$ is one of the essential components in the analysis of the accuracy of Bowen ratio (Allen et al., 2011b). Correlating the LAI with the G/$R_n$ produced a moderate polynomial (3$^{rd}$ order) relationship for both the METRIC ($R^2 = 0.288$ and p>F = 0.23) and the EC ($R^2 = 0.313$ and p>F = 0.15) methods. However, the results suggested that beyond a certain value of LAI (up to 4.2), the relationship between the G/$R_n$ and LAI was decreased. The scatter in the G/$R_n$, where the LAI increased to 4, showed an increasing trend as the values of G/$R_n$ for full canopy cover are ranging between 0.05 and 0.15 as shown in Waters et al. (2002).

### 3.3.3 Sensible heat flux (H)

After the adjustments of EB closure, the error in recorded $H_{EC}$ flux was ranged from 10.89 W m$^{-2}$ (low LAI conditions) to 38.91 W m$^{-2}$ (full canopy condition). It may be due to advection, which varies very strong in the hyper-arid environment. The $H_{METRIC}$ estimated with an MBE value of 15.72 W m$^{-2}$ (13.87 %) compared to $H_{EC}$. The high RMSE value of 72.01 W m$^{-2}$

(63.54 %) for the $H_{METRIC}$ might be due to the selection of anchor pixels especially during 22$^{nd}$ August, where the Landsat-8 image was viewed at the time of irrigation. Hence, the advection and variability in the wetness across the fetch /FTP area and canopy reflectance affected the calculations in RS estimated of $H_{METRIC}$ (Brutsaert and Stricker, 1979; Yang et al., 2014). During the earlier (June) and late summer (August) periods, the observed sharp humidity variations are linked to changes in wind direction. However, during the alfalfa post-harvest practices, the fields are under fallow condition. Hence, the LE flux

was always less as most of the available energy partitioned as H rather than LE flux. This was evident in the linear regression analysis (**Fig. 5**), where a good correlation between the $H_{RS}$ and $H_{EC}$ ($R^2 = 0.61$) was observed; however, it was not significant (p>F = 0.021), and it was also confirmed with the MBE of 13.87 %. The obtained results are resembling with the reported values on $H_{METRIC}$ performance (MBE of 10 %) by Carrasco-Benavides et al. (2013).

### 3.3.4 Latent heat flux (LE)

A scatter plot was established between the METRIC estimated (LE$_{RS}$) and the EC determined LE (LE$_{EC}$) values (**Fig. 5**). The correlation was relatively good ($R^2 = 0.66$); however, it was not statistically significant (p>F = 0.14). This is attributed to the fact that the LE$_{RS}$ is calculated as a residual component of the energy balance equation that mainly depends on the calibration accuracy of the $H_{RS}$. The value of the latter is purely based on the quality of the selected anchor pixels (Weaver, 1990; Field et al., 1994).

Although the regression analysis showed a non-significant correlation, the performance indicators such as the MBE (2.45 W m$^{-2}$; 0.73 %) and the RMSE (115.04 W m$^{-2}$; 34.33 %) are in accordance with the earlier reported values indicating that the METRIC algorithm is rather accurate for determining LE over large areas. Carrasco-Benavides et al. (2013) stated that the METRIC algorithm overestimated the LE by 14 % (RMSE). The obtained average absolute error of the LE$_{RS}$ for the eight images are in our study 35 %. The obtained inaccuracy may be due to the energy partition at peak growth stages, where the

LE accounted for 80-85 % of the net radiation. In addition, the H component accounted only for 5 to 8 % of the Rn during the growing season on the daily scale. For the fallow periods, the recorded partitions of LE and H were in the range of 6.9 to 8.2 % and 77 to 84 %, respectively. On the monthly time scale, H and LE fluxes varied throughout the cropping periods. The METRIC algorithm is therefore observed to perform a relatively better in full canopy conditions compared to that in partial canopy conditions. Due to the cooling effect of vegetation, the selection of anchor pixels for H calculation is challenging in

continuous irrigation regimes.

The LE was high during the full coverage of LAI and irrespective of alfalfa growth stages. Three environmental factors including atmospheric water demand, humidity and wind speed during the growing season have a high impact on the

LE measurements, which in-turn triggered the computation of anchor pixels especially at the time of irrigation. Moreover, in the fallow period, H can be accurately determined with METRIC by selecting the targeted pixels after the harvest of alfalfa. Advection in dry months is common in hyper-arid regions, which affect the H computation, which in-turn affects the LE and ET estimates on a regional and long-term basis.

### 3.3.5 Evapotranspiration (ET)

The correlation between the ET values obtained from both the METRIC algorithm and the EC system is found to be highly significant ($R^2 = 0.93$ and p>F = 0.0001) as illustrated in **Fig. 7**. Further assessment of the accuracy of the METRIC algorithm in estimating the ET is performed using the RMSE and MBE indicators. The performance of METRIC model in estimating ET on hourly and daily interval was significant with the Mann-Whitney U-test (U and Z value are 32 and 0.052 respectively). Similarly, the null hypothesis is not be rejected with Kruskal-Wallis H-test ($\chi^2 = 0.916$). Comparing the hourly ET calculated using the METRIC algorithm to that using the EC system results in an RMSE value of 0.13 mm h$^{-1}$ (25.91 %) and an MBE value of 0.04 mm h$^{-1}$ (6.6 %). For the daily mean ET, the RMSE and MBE values were 4.15 mm d$^{-1}$ (34.33 %) and 0.38 mm d$^{-1}$ (4.2 %), respectively. These results are in agreement with the results reported by Mkhwanazi and Chavez (2012), where the performance errors in estimating the ET using the METRIC model were determined at 0.14 mm h$^{-1}$ (17.6 %) and -0.08 mm h$^{-1}$ (-10.3 %) for RMSE and MBE, respectively. Similarly, Chavez et al. (2007) reported a METRIC error of 0.7 mm d$^{-1}$ (7.4 %) in predicting the ET compared to lysimeter data. On the average, the present study found that the METRIC algorithm overestimated the hourly ET by 6.6 % in comparison to the EC measurements. The daily ET is underestimated by 4.2 % (**Fig. 8**). There is a fluctuation in EC and Landsat estimated Sensible Heat Flux. This might be due to the advection process in case of hourly ET assessment and the variability in the canopy density with respect to the studied footprint. Advection may vary strongly in hyper-arid environments.

### 4 Conclusions

The METRIC algorithm was applied on Landsat-8 images for mapping ET. Its performance was evaluated against the EC flux tower measured energy balance components and ET at hourly and daily intervals over an irrigated alfalfa field in the Eastern Region of Saudi Arabia. The following are the specific conclusions of this study:

- The METRIC algorithm was successful for estimating the ET with an average error of 6.6 % (hourly ET) and 4.2 % (daily ET). The performance of the METRIC algorithm was found to be more accurate in estimating the hourly ET ($R^2 = 0.81$) than the daily ET ($R^2 = 0.66$) compared to the ET$_{EC}$.
- Compared with the EC data, the estimated R$_n$ component from remote sensing was found to be highly accurate (an accuracy of more than 95 %) was obtained. In addition, the H obtained from satellite data was associated with errors ranging between 10.89 W m$^{-2}$ and 38.91 W m$^{-2}$ at low (< 1.2) and high (> 5.0) LAI values, respectively. The corresponding LE estimates have a low mean value of MBE of 2.45 W m$^{-2}$ (0.74 %).

- The METRIC algorithm was observed to perform a relatively better in full canopy conditions compared to that in partial canopy conditions. Due to the cooling effect of vegetation, the selection of anchor pixels for H calculation is challenging in continuous irrigation regimes.

**Acknowledgements**

The authors are grateful to the Deanship of Scientific Research, King Saud University for funding through the Vice Deanship of Scientific Research Chairs. The unstinted cooperation and support extended by Mr. Alan King and Mr. Jack King in carrying out the field research work are gratefully acknowledged. The guidance provided by the subject experts: Biradar, C.M., ICARDA, Jordan and Gowda, P. H., USDA during the study and preparation of the manuscript was quite valuable.

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

**Tables**

**Table 1.** Details of wind parameters at the time of satellite overpass

| Date | Wind Speed (m/s) | | Wind Direction (° from North) | Yaw angle (°) |
|---|---|---|---|---|
| | Min. | Max. | | |
| 03 June, 2013 | 6.02 | 12.45 | 13.54 | 56.46 |
| 19 June, 2013 | 5.61 | 9.69 | 6.49 | 63.51 |
| 05 July, 2013 | 2.43 | 5.28 | 23.98 | 46.02 |
| 21 July, 2013 | 5.25 | 10.71 | 35.13 | 34.87 |
| 22 August, 2013 | 1.61 | 4.91 | 27.80 | 42.20 |
| 07 September, 2013 | 1.11 | 8.17 | 59.82 | 10.18 |
| 23 September, 2013 | 5.12 | 8.93 | 38.13 | 31.87 |
| 09 October, 2013 | 4.60 | 8.67 | 25.40 | 44.60 |

**Table 2.** Energy Balance analysis of EC measured heat fluxes.

| Date | Day of the year 2013 | Stability parameter ($\xi$) | Condition | Bowen Ratio | Energy Balance Closure % |
|---|---|---|---|---|---|
| 3-June | 154 | 0.02 | Stable | -0.17 | 90.84 |
| 19-June | 170 | -0.15 | Unstable | 0.7 | 95.81 |
| 5-July | 186 | 0.07 | Neutral | -0.24 | 99.14 |
| 21-July | 195 | -0.24 | Unstable | 0.63 | 96.80 |
| 22-August | 234 | -0.28 | Unstable | -0.42 | 80.82 |
| 7-September | 250 | 0.04 | Stable | -0.42 | 84.09 |
| 23-September | 266 | -0.03 | Unstable | 0.14 | 75.14 |
| 9-October | 282 | 0.04 | Stable | -0.13 | 71.74 |
| Average | | -0.06 | | 0.01 | 86.79 |

**Table 3.** Details of Landsat-8 data used in the study.

| Path | Row | Date of overpass |
|---|---|---|
| 165 | 43 | June 3, June 19, July 5, July 21, August 22, September 7, September 23 and October 9, 2013 |

**Table 4.** Split-window (SW) coefficient values (Skokovic et al., 2014)

| SW Constant | Value |
|---|---|
| $C_0$ | -0.268 |
| $C_1$ | 1.378 |
| $C_2$ | 0.183 |
| $C_3$ | 54.300 |
| $C_4$ | -2.238 |
| $C_5$ | -129.200 |
| $C_6$ | 16.400 |

**Table 5.** Thermal conversion constants and rescaling factors of Landsat-8 TIRS bands ([a]USGS, 2016; [b]Skokovic et al., 2014)

| Landsat-8 TIRS | Thermal constants[a] | | Rescaling factor[a] | | Emissivity values[b] | |
|---|---|---|---|---|---|---|
| | $K_1$ | $K_2$ | $M_L$ | $A_L$ | $\varepsilon_s$ | $\varepsilon_v$ |
| Band 10 | 1321.08 | 777.89 | 0.000342 | 0.1 | 0.971 | 0.987 |
| Band 11 | 1201.14 | 480.89 | 0.000342 | 0.1 | 0.977 | 0.989 |

**Table 6.** Input parameters used for METRIC model for alfalfa crop.

| DOY | $T_{EC}$ (°C) | $T_{LST}$ (°C) | $LAI_G$ | $ET_r$ | u (m s$^{-1}$) | $h_c$ (m) | $Z_m$ (m) |
|---|---|---|---|---|---|---|---|
| 154 | 36.42 | 34.67 | 1.22 | 1.21 | 0.55 | 0.39 | 3.2 |
| 170 | 39.59 | 33.01 | 3.28 | 0.61 | 0.43 | 0.48 | 3.2 |
| 186 | 34.7 | 32.32 | 4.29 | 0.83 | 0.46 | 0.59 | 2.2 |
| 195 | 40.96 | 32.6 | 5.92 | 0.96 | 0.68 | 0.51 | 3.0 |
| 234 | 42.98 | 40.99 | 1.11 | 0.31 | 0.62 | 0.24 | 3.0 |
| 250 | 35.48 | 32.09 | 3.19 | 0.52 | 0.65 | 0.52 | 3.0 |
| 266 | 34.52 | 34.18 | 1.13 | 0.27 | 0.39 | 0.29 | 2.3 |
| 282 | 26.78 | 24.36 | 2.28 | 0.41 | 0.40 | 0.52 | 3.2 |

DOY – Day of the year; $T_{EC}$ – Temperature from flux tower; $T_{LST}$ – estimated temperature from Landsat-8 images; $LAI_G$ – Ground measured Leaf Area Index; $ET_r$ – Reference ET; u – Average horizontal wind velocity; $h_c$ – alfalfa canopy height; $Z_m$ - Height of wind speed measurement.

**Table 7.** EC measured and METRIC estimated Energy Balance (EB) components over an alfalfa field.

| EB Component | Method | Date of Landsat-8 overpass (Day of the year 2013) | | | | | | | |
|---|---|---|---|---|---|---|---|---|---|
| | | June 03 (154) | June 19 (170) | July 05 (186) | July 21 (195) | Aug. 22 (234) | Sept. 07 (250) | Sept. 23 (266) | Oct. 09 (282) |
| Soil Heat Flux | RS | 28.6 | 92.5 | 143.73 | 80.65 | 97.92 | 48.16 | 19.98 | 98.31 |
| (G), W m$^{-2}$ | ECB | 21.6 | 44.1 | 155.18 | 70.33 | 37.54 | 54.98 | 16.50 | 110.30 |
| Net Radiation | RS | 488.7 | 491.5 | 491.07 | 490.64 | 502.95 | 489.37 | 491.76 | 477.75 |
| (R$_n$), W m$^{-2}$ | ECB | 468.0 | 500.6 | 506.22 | 463.27 | 513.17 | 476.86 | 469.6 | 456.49 |
| G/R$_n$ | RS | 0.06 | 0.19 | 0.29 | 0.16 | 0.19 | 0.10 | 0.04 | 0.21 |
| | ECB | 0.05 | 0.09 | 0.31 | 0.15 | 0.07 | 0.12 | 0.04 | 0.24 |
| Sensible Heat Flux | RS | -57.2 | 184.2 | 286.10 | 160.45 | 1.77 | 95.75 | 39.98 | 195.62 |
| (H), W m$^{-2}$ | EC | -65.8 | 112.3 | 274.59 | 127.34 | 14.26 | 230.79 | 19.03 | 68.39 |
| Latent Heat Flux | RS | 574.5 | 214.8 | 61.25 | 249.55 | 500.20 | 345.46 | 551.72 | 183.82 |
| (LE), W m$^{-2}$ | EC | 335.7 | 160.2 | 72.12 | 260.79 | 637.84 | 505.40 | 560.80 | 148.80 |
| Evapotranspiration | RS | 0.86 | 0.32 | 0.09 | 0.42 | 0.76 | 0.52 | 0.83 | 0.27 |
| (ET), mm hr$^{-1}$ | EC | 0.64 | 0.24 | 0.11 | 0.38 | 0.96 | 0.71 | 0.89 | 0.22 |

RS - Remote Sensing (METRIC algorithm), EC - Eddy Covariance, and ECB - Biomet sensors of Eddy Covariance system.

**Table 8.** Performance indicators results for the METRIC algorithm estimated EB components.

| Error | | Rn W m$^{-2}$ | G W m$^{-2}$ | H W m$^{-2}$ | LE W m$^{-2}$ | ET mm hr$^{-1}$ | ET mm d$^{-1}$ |
|---|---|---|---|---|---|---|---|
| RMSE | Amount | 18.32 | 28.46 | 72.01 | 115.04 | 0.13 | 4.15 |
| | % | 3.74 | 37.33 | 63.54 | 34.33 | 25.91 | 34.33 |
| MBE | Amount | 8.66 | 12.42 | 15.72 | 2.45 | 0.04 | 0.38 |
| | % | 1.76 | 16.29 | 13.87 | 0.73 | 6.6 | 4.2 |
| Nash-Sutcliffe Coefficient | | 1 | 0.59 | 0.59 | 0.88 | 0.99 | 0.99 |
| R$^2$ | | 0.54 | 0.67 | 0.61 | 0.66 | 0.81 | 0.66 |
| P>F | | 0.038 | 0.0131 | 0.0216 | 0.0137 | 0.002 | 0.013 |
| Mann-Whitney U-test# | | 23 (-0.89)* | 26 (-0.58)* | 29 (-0.26)* | 31 (-0.05)** | 32 (0.05)** | 31 (0.05)** |
| Kruskal-Wallis H-test ($\chi^2$) | | 0.893* | 0.397* | 0.099* | 0.011** | 0.021** | 0.001** |

#Z-values are reported in the parenthesis; *Not-significant; **Significant; $\chi^2$ - is the Chi-square

**Figures**

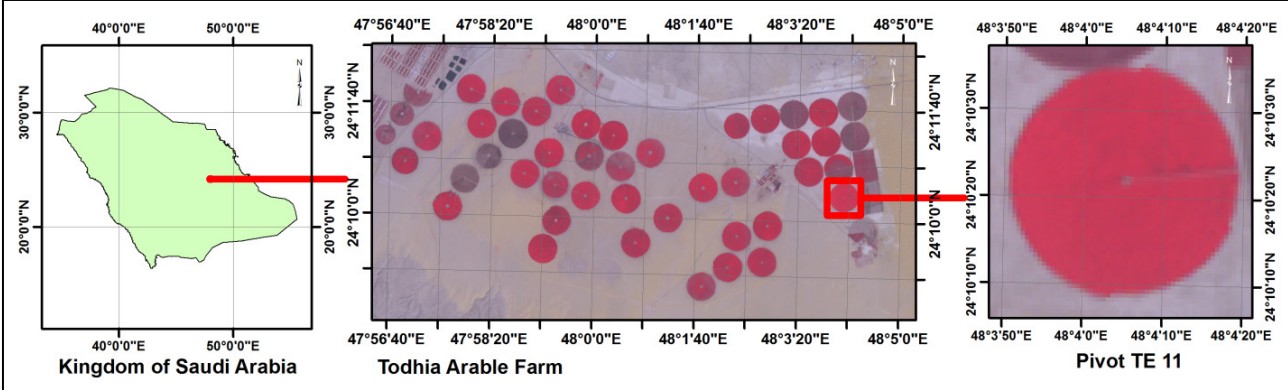

**Figure 1.** Location map of the study area

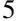5

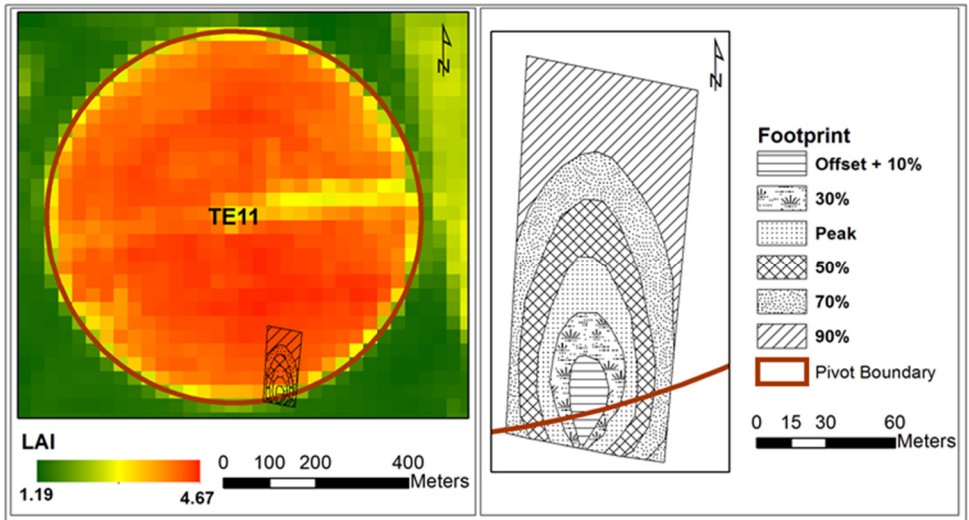

**Figure 2.** Flux Source Area Model (FSAM) Footprint, as % of the fetch area, of the Eddy Covariance system overlaid on LAI map of the alfalfa crop (date of landsat-8 overpass – 21 July, 2013).

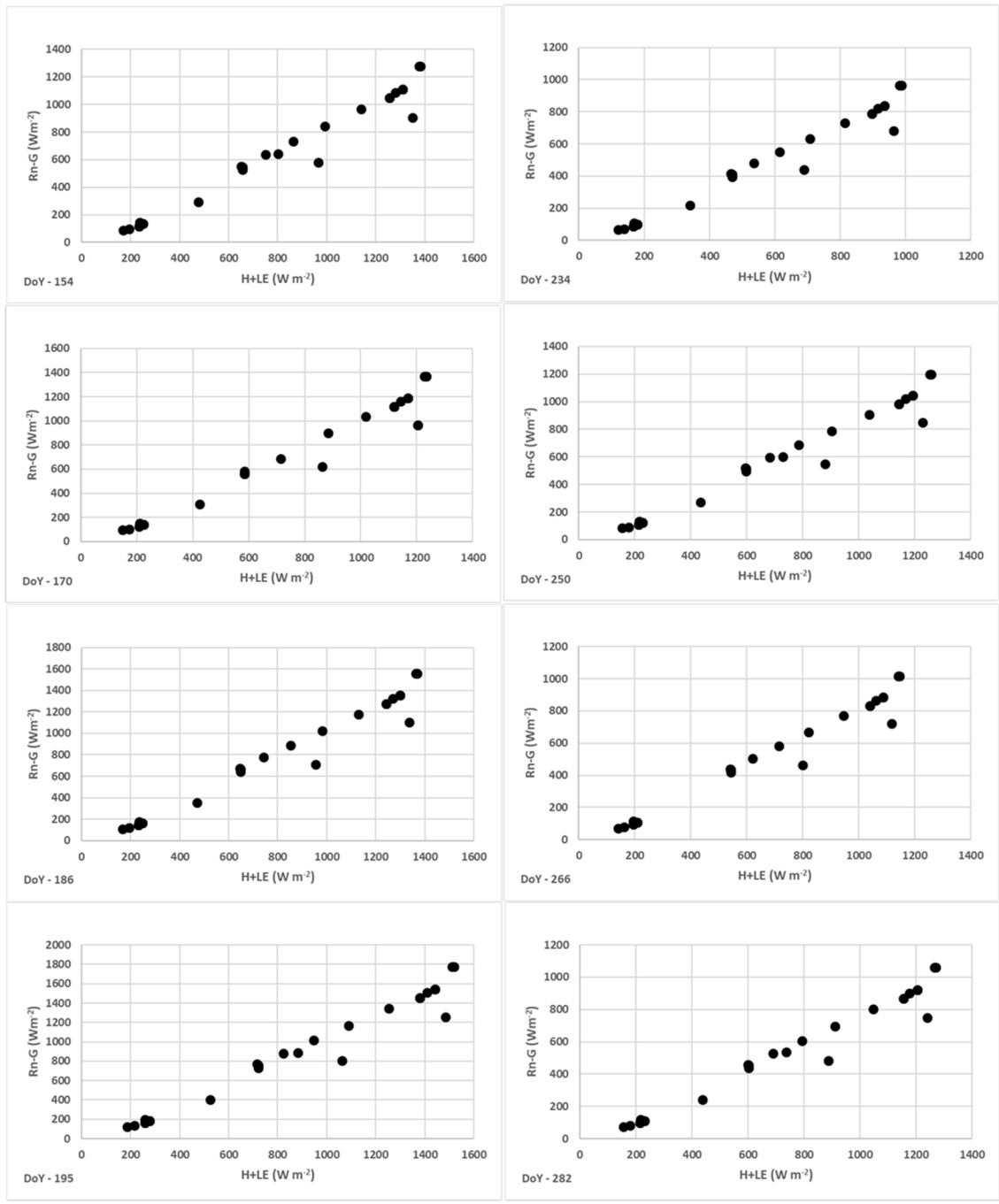

**Figure 3.** Energy balance clousure of corrected Eddy covariance data for the studied landsat-8 date of overpass.

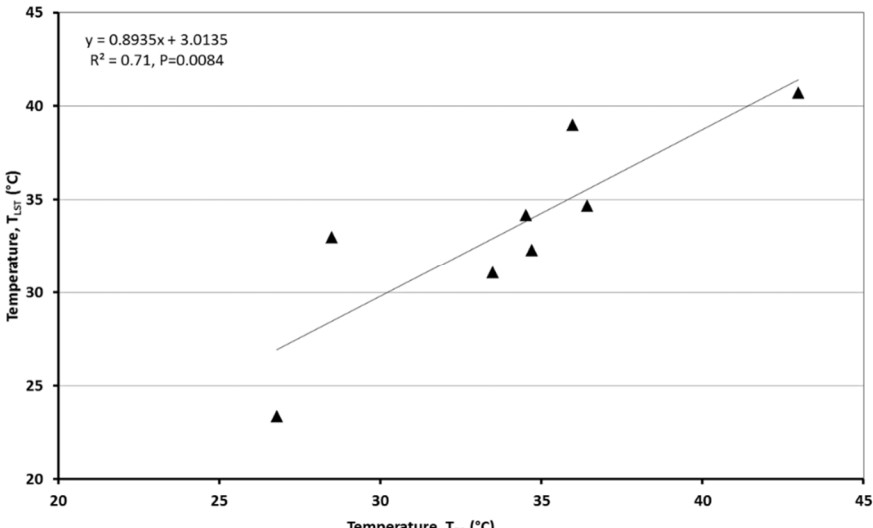

**Figure 4.** Comparison of EC measured ($T_{EC}$) and remote sensing derived ($T_{LST}$) surface temperature.

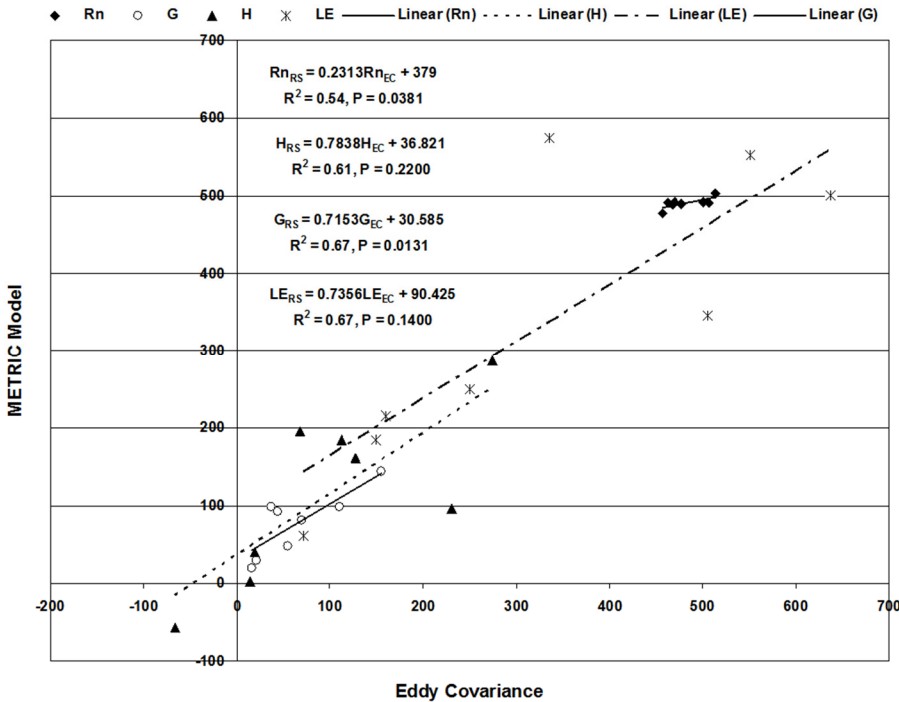

**Figure 5.** The relationship between EB components (W m$^{-2}$) measured by the METRIC algorithm and the EC system.

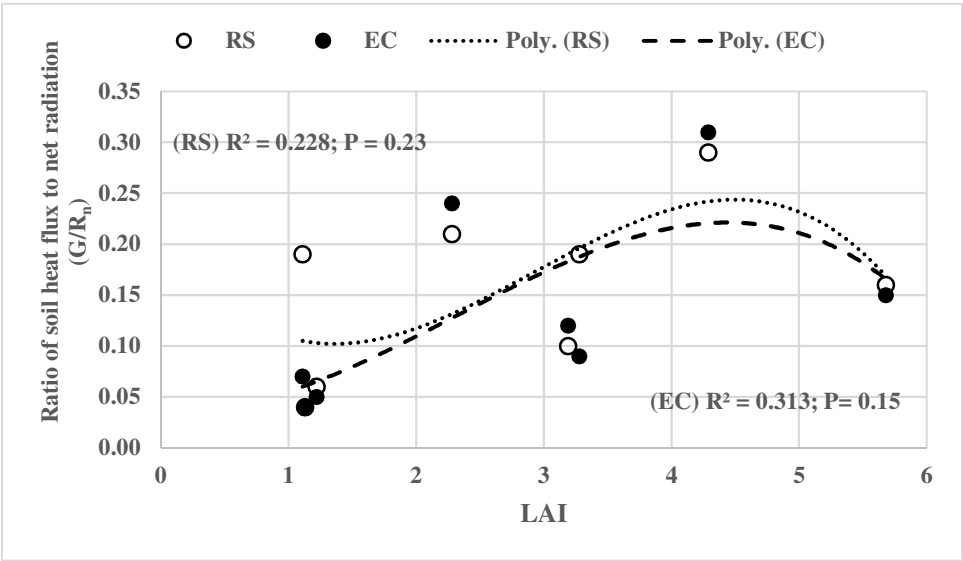

**Figure 6.** LAI and G/Rn relationship of the study crop for remote sensing and Eddy Covariance.

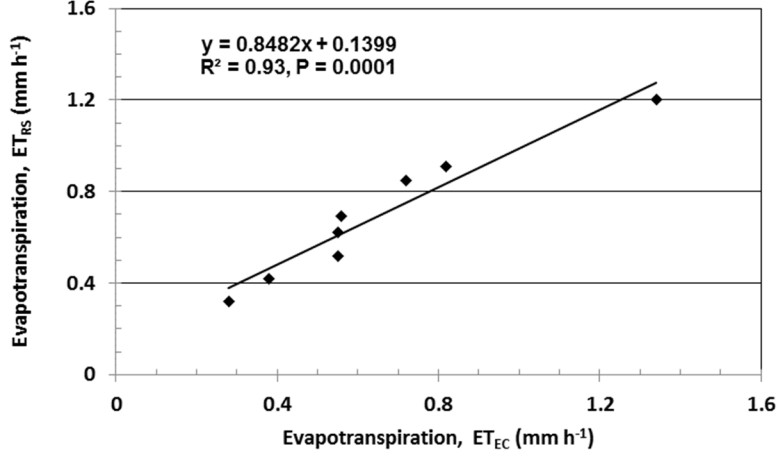

**Figure 7**. METRIC-derived hourly evapotranspiration (ET$_{RS}$) versus Eddy Covariance measurements (ET$_{EC}$).

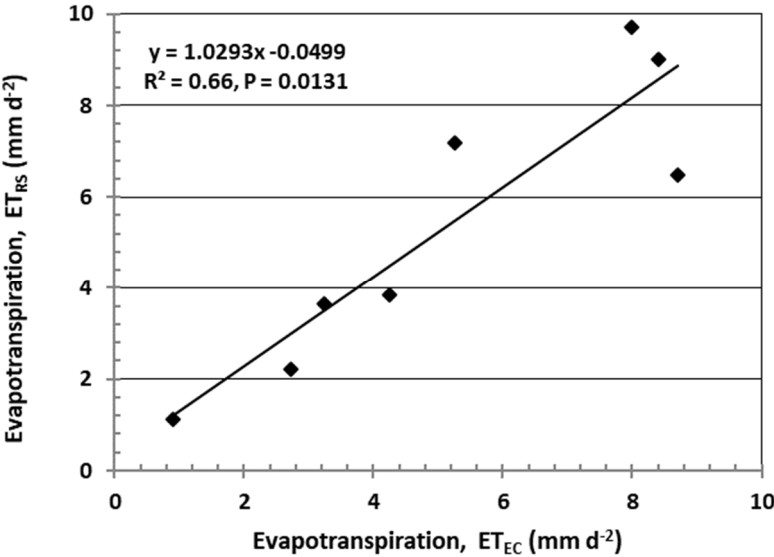

**Figure 8.** The relationship between daily estimates of $ET_{RS}$ and $ET_{EC}$.

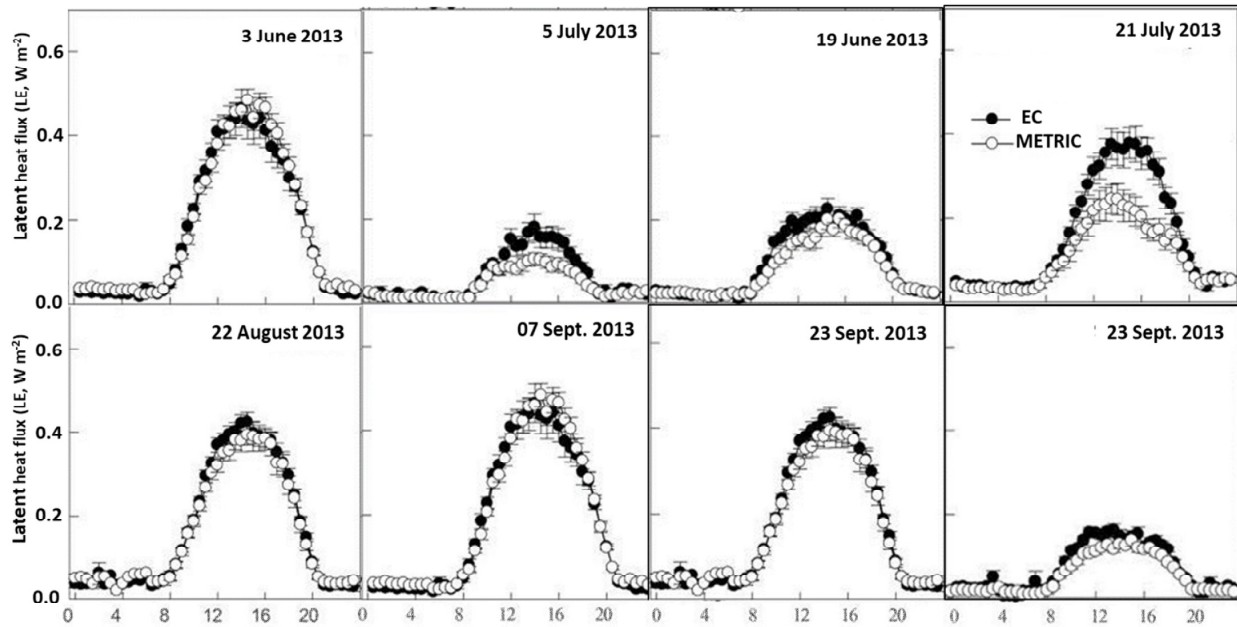

**Figure 9.** Comparative assessment of Eddycovariance fluxtower measured against the METRC estimated Latent Heat flux

10    (LE, W m⁻²)over the study period. (The values are on x-axis are *x1000) data for the studied landsat-8 date of overpass.