# Peer review of "Performance of the METRIC model in estimating evapotranspiration fluxes over an irrigated field in Saudi Arabia using Landsat-8 images"

_Hydrology and Earth System Sciences, 2017_

## Referee Comment (RC1) · Anonymous Referee #1 · 7 Jul 2017

The manuscript entitled "Performance of METRIC in estimating hourly and daily evapotranspiration fluxes over an irrigated field in Saudi Arabia" by Madugundu et al presents the results of a validation study of the METRIC evapotranspiration and surface heat fluxes retrieval algorithm using Landsat-8 satellite images over an irrigated alfalfa field in Saudi Arabia equipped with eddy covariance system units. Although the goal of the study is not clearly stated, to my knowledge, the study is the first of its kind in Saudi Arabia, applying and validating a remote sensing algorithm for surface energy balance over irrigated fields in hyper-arid environment. As such, the study may be interesting to HESS readership interested in the surface energy balance in such environments and in remote sensing algorithms. However, the manuscript 1) lacks crucial information: a

clear stated goal, some clear interpretation of discrepancies, graphics and definition of variables, 2) there are apparent inconsistencies between the scores presented in the abstract, text and tables, 3) some methods are not described (eg FTP) and use of correlation for such small sample of data may lead to not meaningful results. I list here detailed comments I have while reading the manuscript, also including the 3 points mentioned above.

- The title should be revised to explicitly mention the use of satellite remote sensing (Landsat).

- Introduction: the goal seems to be stated in p.3 l. 11-15. However, the three successive sentences are not linked by logical reasoning. The first states about crop water management, which is a matter of almost "real-time" data. The second is about lack long-term spatial data (by the way, there is a spatial ET dataset over part of Saudi Arabia for 1992-2014 processed with SEBAL & MODIS (Mahmoud & Alazba, 2016, J Asian Earth Sciences, 124, 269-283)), which seems not to be connected to the first sentence and do not justify the developments presented in the third sentence. A clear statement of the objective of the study with its motivation and a stress on novelty is needed at the end of the introduction.

- section 2.2: Are the EC data corrected as suggested in the introduction p.2 l.15-25? If not, this need justification. If it is, the method should be described.

- section 2.6 and 2.3: the method for footprint analysis should be described and properly referenced, a name of a program can also be written.

- section 2.6: the use of some coefficients for comparison with such a small sample of observations may lead to not significant conclusions.

- section 3.1: the "tower measured temperature", $T\_EC$, is not defined. What is the height of measurement? If it is not a surface temperature, there is no reason to find an agreement with Landsat land surface temperature, because those relates to different heights. Moreover, the explanation given for the discrepancies is not clear to me. Revisit of the text is probably needed here.

- section 3.2.2: the explanation given in p 8, l.29-31 is not clear. I do not see a constant relation between the two variables from Figure 6 for LAI>4.2, neither is obvious the scatter for LAI>4, as there are only 2 data.

- section 3.2.5 should contain all the statistical results obtained for ET hourly and daily. It is stated in the abstract that hourly ET was overestimated and daily ET underestimated. It should be written in the section.

- there is an inconsistency between the statistical scores for ET given in the abstract, in section 3.2.5, in Table 5, and conclusion: this needs to be clarified.

- the location of section 3.2.6 at the end of the paper is awkward, this should be moved in section 2.2 and clearly linked to the rest of the manuscript.

- Abstract: is there any reason for daily ET to understimate, while hourly ET is overs-estimated by METRIC ?

- Figure 4,5 7 and 8 are already included in figure 9, they could be removed.

- A graphic for daily ET comparison is lacking in the manuscript.

- Table 3: why is Zm changing from date to date? Is a correction done to account for this variation in the model?

---

## Referee Comment (RC2) · Anonymous Referee #2 · 9 Jul 2017

General comments

The study presents an evaluation of the METRIC model on a alfalfa field located in Saudi Arabia. Although the work is not particularly original, the validation could be of particular interest for HESS readers because of the study area located in Saudi Arabia that is not well known. Nevertheless, I have some major concerns: (1) the reliability of the results: there are many discrepancies in the provided statistical values between the abstract, text, figures and tables; (2) the methodology lacks important information and (3) the results discussion should be considerably strengthened.

Specific comments

[Figure]

*The introduction is poorly structured: L.15-25: The paragraph is described in vague terms. The conclusion "the ET values measured by the EC system need to be adjusted, through an appropriate method, to improve their accuracy" is not clear to me. What the authors want to explain ? That the energy closure of EC system is not always satisfied ? I am not sure that an entire paragraph should be devoted to this point. Twine et al. (2000) should be cited then. Twine, T.E., Kustas, W.P., Norman, J.M., Cook, D.R., Houser, P.R., Meyers, T.P., Prueger, J.H., Starks, P.J., Wesely, M.L., 2000. Correcting eddy-covariance flux underestimates over a grassland. Agric. For. Meteorol. 103, 279–300. doi:10.1016/S0168-1923(00)00123-4

The state of the art concerning the RS approaches to monitor spatialized ET is not sufficiently detailed. The FAO-56 approach is an interesting alternative to thermal based approach and thermal based approaches are usually separated into image-based method (named contextual) and pixel to pixel based where the energy balance is solved independently from one pixel to another. The cited article Kalma et al., 2008 together with Courault et al., 2005 could certainly help to improve the introduction.

Courault, D., Seguin, B., Olioso, A., 2005. Review on estimation of evapotranspiration from remote sensing data: From empirical to numerical modeling approaches. Irrig. Drain. Syst. 19, 223–249. doi:10.1007/s10795-005-5186-0

The objectives are not clearly stated.

*The study area describtion should be strengthen. Please provide some details on the typical annual cycle of alfafa crop in the region (in the 2.1 part for instance) and on the soil type

*LANDSAT8 LST: please provide some details on the split windows algorithm and give the proper references of the software.

*P7 L1-7: give some detail on the Footprint analysis approach

*P.7 L23: What is the "EC flux tower measured temperature (TEC)" ? Is it derive from

the upward longwave component measured by the CNR4 ?

*The discussion on the results should be strenghten: - Providing scatterplot only does not help in this objective. A time series, of at least LE, showing both in situ and satellite estimates should be shown and discussed

- Discussing on statistics with such a small sample of data may be uncertain.

- Please organize and strenghten your discussion. For instance, the "Sensible heat flux" part (3.2.3) is very difficult to follow after the first sentence where you provide the statistics of the comparison between EC and metrics:

"The high RMSE value of 72.01 W m-2 (63.54%) for the HRS might be due to the advection and variability in the canopy density."

Right. Advection may very very strong in hyper arid environment but you could give some references to support your comment.

"Hence, most of the Rn has been partitioned into LE than into H, as introduced by the near surface air temperature difference ($\Delta$T) and the aerodynamic resistance of heat transfer (rah), i.e. propagation errors."

Not clear to me.

"This was evident in the linear regression analysis (Figure 7), where a good correlation between the HRS and HEC (R2 = 0.61) was observed; however, it was not significant (P>F = 0.022), and it was also confirmed with the RMSE 10 of 63.5%."

I don't the see the link with the preceding sentence.

"In contrast, Carrasco-Benavides et al. (2013) reported that the METRIC algorithm overestimated the H component by 39 W m-2 with a mean absolute error of 10%."

In the previous sentence, you were commenting the correlation and the RMSE, this one refers to bias. Not clear.

* Energy balance (3.2.6): please provide a figure of the EC EB closure in the section 2.2 (and explain if a correction for EB closure has been applied). The discussion on the EB closure at the date of the LANDSAT images acquisition should be put earlier in the results section

*Please check the consistency of the statistical value in the abstract, text, table and figures.

Technical corrections

p.1 L. 29: please replace "can" by "could" p.2 L12-14: should not be placed here just after the listing of in situ approaches to monitor ET (FAO p2. L20: "techniques of energy closure and BR" Bowen ratio is a technique of energy closure here p2. L24: "the accuracy from 79.2% to 95.2%." replace "accuracy" by "closure" p.3 L11-12: already written earlier in the introduction p.3 L22-23: "Due to the high crop water demand combined with the highly erratic rainfall, irrigation is entirely provided" to be replaced by "Due to . . . ., irrigation is a pre-requisite for crop growth. It is entirely provided by . . ." p.4 L6: "The missed data was filled" please provide some details on filling the missing data p.5 L11-12: to be put in the part describing the pre-processing of LANDSAT8 data P.6 L31: ET24 already described above P7 L9: check the numbering for all part. p.7 L26-27: Please reformulate as lysimeters does not provide any measurement of surface temperature

---

## Author Comment (AC1) · 22 Aug 2017

Author Response to Anonymous Referee #1

Referee comment (RC); Author Response (AR)

RC-1: Lack crucial information: a clear stated goal, some clear interpretation of discrepancies, graphs, and definition of variables.

AR-1: As suggested, crucial information such as (i) clear statement of goal, (ii) strengthening of results and discussion part with-out any discrepancies, and (ii) presentation, graphs, and definition of variables, has been modified.

RC-2: There are apparent inconsistancies between the scores presented in the abstract, text and tables.

AR-2: Agreed, there were an inconsistency between the scores presented in the abstract, text and tables due to oversite at the time of compilation/manuscript preparation. The discrepancies have been rectified.

RC-3: Some methods are not discribed (eg. FTP) and use of correlation for small samples of data may lead to not meaningful results.

AR-3: Methods section will be modified as per referee comments. A detailed description on FTP method will be provided in the revised manuscript. As the referee raised that "the use of correlation for small samples of data may lead to not meaningful results". Results obtained with the use of correlation procedure will be given less importance. Alternatively, Mann-Whitney U-test and/or Kruskal-Wallis H test, which are often used in applications involving small size samples, will be used as described in Gisondi et al. (2004) and McCune and Grace (2002).

RC-4: - The title should be revised to explicitily mention the use of satellite remote sensing (Landsat-8).

AR-4: As suggested, the title has been modified, "Performance of METRIC Model in estimating Evapotranspiration fluxes over an irrigated filed in Saudi Arabia with the use of Landsat-8 images".

RC-5: - Introduction: the goal seems to be stated in p.3 L.11-15, however, the three succssive sentences are not linked by logical reasoning. The first states about crop water management, which is a matter of almost "realtime" data. The second is about lack long-term spatial data (by the way, there is a spatial ET dataset over part of saudi arabia procssed for 1992-2014 processed with SEBAL and MODIS (Mahmoud & Alazba, 2016, J Asian Earth Sciences, 124, 269-283), which seems not to be connected to the first scntence and do not justify the developments in the third sentence. A clear state-

ment of the objective of the study with its motivation and a stress on novelty is needed at the end of the introduction.

AR-5: As suggested, the goal of the study has been stated clearly in the revised manuscript. The sentences were logically linked. A clear statement of the objective of the study with its motivation is added.

RC-6: Section 2.2: Are the EC data corrected as suggested in the introduction p.2 L. 15-25? If not, this needs justification. If it is, the method should be described.

AR-6: Yes, the EC data was corrected as stated in the introduction. As suggested, text on methods/techniques used in each step of EC data correction has been added in the revised manuscript.

RC-7: Section 2.6 and 2.3: the method for footprint analysis should be described and properly referenced, a name of a program can also be written.

AR-7: As suggested, detailed description on "footprint analysis model" used for the study has been added.

RC-8: Section 2.6: the use of some coefficients for comparision with such a small sample of observation may lead not to ignificant conclusions.

AR-8: Results obtained with the use of correlation procedure will be given less importance. Alternatively, MRPP procedures (Mann-Whitney U-test and/or Kruskal-Wallis H test), which are often used in applications involving small size samples will be used as described in Gisondi et al. (2004) and McCune and Grace (2002).

RC-9: section 3.1: the "tower measured temperature", T_EC, is not defined. What is the height of measurement? If it is not a surface temperature, there is no reason to find an agreement with Landsat surface, because those relates to different heights. Moreover, the explanation given for the discrepancies is not clear to me. Revise of the text is properly needed here.

[Figure]

AR-9: The term "tower measured temperature", T_EC, has been defined. The height of the measurement is 3.74 m from the soil surface. It is a surface temperature and can be used in the comparative assessment of Landsat estimated surface temperature. The inconsistencies in the explanation has been removed and the text has been modified.

RC-10: Section 3.3.2: the explanation given in p 8, L 29-30 is not clear. I do not see a constant relation between the two varibles from Figure 6 for LAI>4.2, neither is obvious the scatter for LAI>4, as there are only 2 data.

AR-10: The explanation pertaining to LAI has been revised with clear statements.

RC-11: Section 3.2.5 should contain all the statistical results obtained for ET hourly and daily. It is stated in the abstract that hourly ET was overestimated and the daily ET underestimated. It should be written in this section.

AR-11: As suggested, all the statistical results obtained for ET (hourly and daily) is stated and the section has been modified accordingly.

RC-12: There is an inconsistancy between the statistical scores for ET given in the abstract, in section 3.2.5, in Table, and conclusion: this needs to be clarified.

AR-12: All the datasets, obtained results and their analysis has been reviewed. Inconsistency among statistical scores across the manuscript has been corrected and the concerned sections/parts have been modified accordingly.

RC-13: The location of section 3.6 at the end of the paper is awkrard, this should be moved to section 2.2 and clearly linked to the rest of the manuscript.

AR-13: As suggested, section 3.6 has been merged with section 2.2 and linked to the rest of the manuscript.

RC-14: Abstract: is there any reason for daily ET to underestimate, while hourly ET is overestimated by MERIC?

AR-14: There is a fluctuation in EC and Landsat estimated Sensible Heat flux, it might

be due to the advection in case of hourly ET and variability in the canopy density with respect to studied footprint". Advection may vary strongly in hyper-arid environments.

RC-15: Figure 4,5,7 and 8 are already included in Figure 9, they could be removed.

AR-15: As suggested, Figure 4, 5, 7 and 8 has been removed in the revised manuscript.

RC-16: A graphic for daily ET comparision is lacking in the manuscriept.

AR-16: Graphical representation of daily ET (EC measured and METRIC estimated) will be provided with the revised manuscript.

RC-17: Table 3: why is Zm changing from date to date? Is a correction done to account for this variation in the model?

AR-17: Depending on the height of crop, the Zm was varied. Yes, the correction was done with the use of an internal component of Eddypro software.

---

## Author Comment (AC2) · 22 Aug 2017

Author Response to Anonymous Referee #2

Referee comment (RC); Author Response (AR)

General comments:

RC-1: The reliability of results: there are many discrepancies in the provided statistical values between the abstract, text, figures and tables.

AR-1: Agreed, there were discrepancies in provided statistical values among the abstract, figures and tables. Those discrepancies have been rectified and the corresponding section of the manuscript has been revised.

RC-2: The methodology lacks important information.

AR-2: Detailed description of methods used for the study has been incorporated.

RC-3: The results discussion should be considerably strengthened.

AR-3: The results and discussion sections have been modified considerably.

Specific comments:

RC-4: - The introduction is poorly structured:L 15-25:The paragraph is described in vague terms. The conclusion "the ET values measured by the EC systems need to be adjusted through, through an appropriate method, to improve their accuracy" is not clear to me. What the author want to explain? The energy closure of EC system is not always satisfied?, I am not sure that an entire paragraph should be devoted to this point. Twine et al. (2000) should be cited then.

AR-4: Introduction and conclusion sections have been modified. The explanation on Energy closure of EC system modified and the Reviewer suggested references were incorporated in the revised manuscript.

RC-5: - The state of art concerning the RS approaches to monitor spatialized ET is not sufficiently detailed. The FAO-56 approach is an interesting alternative to thermal based approach and thermal based approaches are usually separated into image-basedmethod (named contextual) and pixl to pixel based where the energy balance is solved independently from one pixel to another. The sited article Kalma et al., 2008 together with Courault et al. 2005 could be certinly help to improve the introduction.

AR-5: The image-based method has been followed for the monitoring of spatialized ET. As suggested, reference of Courault et al. (2005) is added along with Kalma et al. (2008).

RC-6: The objectives are not clearly stated.

**HESSD**

AR-6: The goal and objectives of the study have been stated clearly in the revised manuscript.

RC-7: The study area description should be strengthened. Please provide some details on the typical annucal cycle of alfalfa crop in the region (in the 2.1 part of instance) and on th soil type.

AR-7: Description of the study area, the typical annual cycle of alfalfa crop in the region and soil type of experimental plot has been added.

RC-8: Landsat8 LST: Please provide some details on the split window algorithm and give the proper references of the software.

AR-8: A detailed explanation on "split window algorithm" has been provided with the proper reference to the software. ENVI (Ver. 5.1) software has been used for the execution of split window model.

RC-9: P7 L1-7: give some detail on the Footprint analysis approach

AR-9: A detailed text on Footprint analysis approach is added to the revised manuscript.

RC-10: P7 L23: what is the "EC flux tower measured temperature (TEC)"? is it dereived from upward longwave component measured by the CNR4?

AR-10: Explanation pertaining to TEC and its method of measurement is provided in the revised manuscript. Yes, the TEC is the product of a series of computations from upward longwave component measured by the CNR4.

RC-11: The discussion on the results should be strengthened: - Providing scatter plot only doesnot help in this objective. A time-series, of at least LE, showing both insitu and satellite estimates should be shown and discussed.

AR-11: Discussion and results sections has been strengthened. A time-series of LE showing both in-situ and satellite (landsat8) estimates has been provided and discussed accordingly.

RC-12: Discusing on statistics with such small sample data may be uncertain.

AR-12: Results obtained with the use of correlation procedure will be given less importance. In addition, Mann-Whitney U-test and/or Kruskal-Wallis H test, which are often used in applications involving small size samples will be used as described in Gisondi et al. (2004) and McCune and Grace (2002).

RC-13: Please organize and strengthen your discussion. For instance, the "Sesible heat flux" part (3.2.3) is very difficult tofollow ater the first sentence whereyou provide the statistics of the comparision between EC and metrics.

AR-13: Discussion and results sections will be strengthened.

RC-14: The high RMSE value of 72.01 W m-2 (63.54%) for the HRS might be due to the advection and variability in the canopy density". Right. Advection may vary strong in hyper arid environment but you could give some references to support your comment.

AR-14: As suggested, appropriate references were provided to support the given statements.

RC-15: Hence, most of the Rn has been partitioned into LE than into H, as introducedbythenear surfce air temperature differenc ($\Delta$T) and the aerodynamic resistance (rah), i.e. propagation errors". Not cler to me

AR-15: More explanation is provided and the discrepancies in the statement has been removed.

RC-16: "This was evident in the linear regression analyssis (Figure 7), . . . . . . . . . . . . . . with the RMSE 10 of 63.5%". I don't see the link with the preceding sentence."In contrast,Carrasco-enavids etal. (2013) . . . . . . . error of 10%". In the previous sentence, you are commenting the correlation and the RMSE, this one referes tobias, Not clear.

AR-16: As suggested, appropriate references were provided to support the given statements.

RC-17: Energy balance (3.2.6): Please provide a figure of EC EB clouer in the section 2.2 (and explain if a correlation for EB clouer has been applied). The discussion on the EB clouser at the date of the LANDSAT images acquisition should be put earlier in the results section.

AR-17: More explanation is provided on Energy Balance Closure. Discrepancies in the statement has been removed. The EB closure and the date of the Landsat image acquisition will be provided prior to the results section.

RC-18: Please check the consistancy of the statistical value in the abstract, table and figures.

AR-18: Agreed, there were discrepancies in provided statistical values among the abstract, figures and tables. Those discrepancies have been rectified and the corresponding section of the manuscript has been revised.

RC-19: Technical corrections.

AR-19: All suggested technical corrections has been incorporated in the revised manuscript.

---

## Author Response (AR2)

**Author response to Reviewer comments**

**RC: Reviewer comments; AR: Author response to reviewer comments**

RC-1: Page 3, Line 3: "Add reference".

AR-1: A suitable reference was added in the revised manuscript (Page: 2, Line: 34)

RC-2: Page 3, Line 16-18: "I think that the formulation of the objectives should still be sharpened. What is the exact ambition as compared to the state of the art".

AR-2: The objectives have been fine-tuned (Page:3, Line: 13-16 )

RC-3: Page 3, Line 27: "Add scientific names of the crop"

AR-3: Scientific names of crops were provided in the revised MS (Page: 3, Line: 25-26)

RC-4: Page 8, Line 10: "You should explain how Rs_down, Rl_down and Rl_up have been assessed. Directly from landsat band? In other words how are LST values and L lambda linked to this!"

AR-4: Computation of RS_down, RL_down and RL_up has been explained in the revised MS (page: 8, Line: 10 to Page 10, Line:9). The utilization of LST and L_lambda in the calculation of energy flux components was also explained (Page: 6, Line: 24-25) and the LST was utilized in Equation 16 and 19 (Page: 9).

RC-5: Page: 11, Line: 2-3: Does this apply for the Omran et al., study?

AR-5: No, it does not belong to Omran et al., study – The ambiguity in the sentence has been removed (Page: 12 Line: 19-20).

RC-6: Page: 11, Line: 4-5: This is very unclear. Should be rephrased.

AR-6: As suggested, the sentence has been rephrased (Page: 19   Line: 20-22).

RC-7: Page: 11, Line: 8: The low and high LAI cases.

AR-7: The sentence has been modified (Page: 12, Line: 26-27).

RC-8: English edition, grammar check and fine-tuning of the manuscript.

AR-8: As suggested by the reviewer, the MS has been subjected to English grammar, edition.

---

## Author Response (AR3)

**Author response to Editor Comments (EC)**

Editor Comment: The manuscript is now nearly ready for publication, but needs further English improvement.
Author Response: The manuscript has been checked for grammar and editions.

[revised manuscript text omitted]